# Association of a Priori-Defined Dietary Patterns with Anthropometric Measurements: A Cross-Sectional Study in Mexican Women

**DOI:** 10.3390/nu11030603

**Published:** 2019-03-12

**Authors:** Mohammad Sediq Sahrai, Inge Huybrechts, Carine Biessy, Marc James Gunter, Isabelle Romieu, Gabriela Torres-Mejía, Laure Dossus

**Affiliations:** 1Nutrition and Metabolism Section, International Agency for Research on Cancer (IARC/WHO), 150 cours Albert Thomas, 69372 Lyon CEDEX 08, France; sahraiM@students.iarc.fr (M.S.S.); HuybrechtsI@iarc.fr (I.H.); biessyc@iarc.fr (C.B.); gunterm@iarc.fr (M.J.G.); 2National Institute of Public Health, Centre for Population Health Research, Universidad No. 655 Colonia Santa María Ahuacatitlán, Cerrada Los Pinos y Caminera C.P., Cuernavaca 62100, Morelos, México; iromieu@gmail.com (I.R.); gtorres@insp.mx (G.T.-M.)

**Keywords:** obesity, overweight, a priori dietary patterns, Mexican women

## Abstract

This cross-sectional study aimed to evaluate associations between a priori defined dietary patterns and anthropometric measures in Mexican women. A total of 1062 women aged 35 to 69 years old from the control participants of the CAMA (Cancer de Mama) study, a multi-center population-based case-control study on breast cancer conducted in Mexico, were interviewed and dietary intakes were assessed using questionnaires. The following indices were derived from these data: Dietary Approaches to Stop Hypertension (DASH) score, the Healthy Eating Index (HEI), the Mediterranean Diet Score (aMED), the Diet Quality Index (DQI), glycemic index (GI) and glycemic load (GL). Adjusting for age, center, educational level, physical activity and energy intake, a high GI was positively associated with a higher body mass index (BMI) and waist circumference (WC). Higher adherence to aMED was associated with lower WC and waist-to-hip ratio (WHR) but no significant association was observed with other a priori dietary patterns. In this population of Mexican women, higher adherence to Mediterranean diet was associated with lower WC but other a priori dietary scores appeared to be of limited value in exploring the association between diet and anthropometric measures.

## 1. Introduction

The prevalence of obesity, a preventable risk factor for a number of non-communicable diseases (NCD), has nearly tripled worldwide since 1975. In 2016, 1.9 billion adults (18 years and older) were overweight (39%) and over 650 million (13%) were obese [1]. In Mexico in 2012, the prevalence of overweight and obesity reached 38.8% and 32.4% respectively [2] and Mexico is among the countries with the highest prevalence of overweight and obesity. The role of diet and its components in the aetiology of overweight and obesity has been extensively studied and adherence to a healthy diet, characterized by increased intakes of nutrient-dense foods like whole grains, fruits and vegetables and reduced consumption of energy-dense fast foods and desserts, has been shown to prevent obesity [3].

Dietary patterns are used to assess overall diet quality as an alternative to the traditional approach, where food groups or single nutrients were examined as exposures for evaluating the association between diet and health [4]. A systematic assessment of dietary patterns across 187 countries showed that diet quality varied among different world regions by age, sex, time, and national income, where high and middle income countries were improving their diet quality between 1990 and 2010 (higher consumption of healthy items and/or lower consumption of unhealthy items) but no improvement was observed in low income regions [5]. Most countries in the developing world have experienced dietary changes in the form of an increasing consumption of fats and added sugar in their diet, while intake of cereals and fiber has declined [6]. One approach to evaluate dietary patterns is to use a priori scores to assess the overall diet quality, based on dietary recommendations. Different a priori dietary indices such as Dietary Approaches to Stop Hypertension (DASH) score [7], the Healthy Eating Index (HEI) [8], the Mediterranean Diet Score (aMED) [9], and the Diet Quality Index (DQI) [10], have been proposed to be associated with a decreased risk of chronic diseases.

In 2014, the United States Department of Agriculture (USDA) performed a systematic review on the relationship between dietary indexes (including those cited above) and health outcomes [11]. Based on results from 14 studies all conducted in high income countries (HICs), they concluded that adherence to scores high in fruits, vegetables, whole grains, legumes, unsaturated oils, and fish, while low in total meat, saturated fat, cholesterol, sugar-sweetened foods and beverages, and sodium, and moderate in dairy products and alcohol, such as aMED, DQI, and HEI, were associated with a decreased risk of obesity. It has also been proposed that high glycemic index (GI) and glycemic load (GL) are associated with an increased risk of obesity and their role in the prevention of overweight, obesity, and other chronic diseases was broadly investigated [12].

These dietary patterns which are based on a priori indices have been mostly developed from studies on Caucasian populations and it is less known whether this approach is appropriate for Hispanic populations, specifically Mexicans, whose adherence to these indexes is hypothesized to be low as described below [13].

A report based on a national health and nutrition survey conducted in 2012 in Mexico showed that the dietary quality of the Mexican population is poor and that a majority of the population does not meet dietary recommendations, with excessive intake of added sugars and saturated fats and low intake of fruits, vegetables, legumes, or seafood [14]. However, to our knowledge, only one study has been conducted on the relationship between a priori dietary patterns and obesity in the Mexican population, where characteristics and quality of a Mexican diet were assessed using a cardioprotective index (CPI), a micronutrient adequacy index (MAI), and a dietary diversity index (DDI) [15]. Therefore, we wanted to evaluate the associations of consumption of a priori dietary patterns with anthropometric measures in a group of 1074 controls from the CAMA (Cancer de Mama) study. The objective of our study was to evaluate the associations between a priori defined dietary patterns, including DASH (Dietary Approach to Stop Hypertension), DQI (Diet Quality Index), aMED (Mediterranean Diet Score), HEI (Healthy Eating Index), Glycaemic Index (GI), and Glycaemic Load (GL) and various anthropometric measures in Mexican women.

## 2. Materials and Methods

### 2.1. Study Population

The study population was selected from the control subjects of CAMA (Cancer (CA) de mama (MA)) study. The rationale and design of the CAMA study has been described in detail elsewhere [16]. In brief, CAMA was a multi-center population-based breast cancer case-control study conducted by the National Institute of Public Health in Cuernavaca, Mexico. All participants were women aged between 35 and 69 years. One thousand newly diagnosed breast cancer cases and 1074 controls were recruited for the study. Cases were identified by professional nurses at different hospitals from medical records and pathology reports. Controls were randomly selected by multiple-step random sampling and were frequency-matched to cases according to 5-year age groups, health-care system and place of residence. The enrollment of all participants took place between 2004 and 2007. For the current analyses, 12 control participants with missing dietary data were excluded, resulting in a study population of 1062 cancer-free women. This study was conducted according to the guidelines laid down in the Declaration of Helsinki and all procedures involving human subjects were approved by the Institutional Review Board at the National Institute of Public Health. Written informed consent was obtained from all subjects.

### 2.2. Data Collection

Participants were interviewed by the study personnel at recruitment and they all attended the hospital for anthropometric measures and mammography screening and to provide fasting blood samples.

#### 2.2.1. General Questionnaire Including Socio-Economical and Health Information about the Participant

Each participant answered questions about sociodemographic characteristics; personal and familial medical history (chronic diseases such as hypertension, diabetes mellitus); gynecological and obstetric history (e.g., age at menarche and menopause, number of pregnancies, duration of lactation, intake of oral contraceptives and hormones); diet; lifestyle factors (e.g., smoking, physical activity). More details on the type of data collected have been published in [16,17,18,19,20].

#### 2.2.2. Anthropometric Measurements

Anthropometric indices were measured by trained nurses. Body weight of the participants was measured at nearest 0.1 kg using a digital electronic scale (Tanita). Height was measured in standing position to the nearest millimeter with a stadiometer (SECA, Hamburg, Germany). Circumference of the waist (WC) was measured while sitting at the umbilicus level and hip circumference in standing position at the level of the most prominent part of the gluteus. Body mass index (BMI) was calculated as weight (kg) divided by height (in meters) squared. Waist-to-hip ratio (WHR) was calculated based on waist (in centimeters) divided by a hip circumference (in centimeters).

#### 2.2.3. Dietary Intake Assessment

Dietary intake information was collected by asking the participant about her food consumption during the last year, using a semi-quantitative Food Frequency Questionnaire (FFQ). This questionnaire was adapted from a validated FFQ developed by Willett [21], considering the Mexican eating culture and was validated among Mexican women [22]. The FFQ contained 104 items and ten multiple-choice frequency categories of consumption: ‘6 or more per day’, ‘4–5 per day’, ‘2–3 per day’, ‘1 per day’, ‘5–6 per week’, ‘2–4 per week’, ‘1 per week’, ‘1–3 per month’, ‘less than 1 per month’, and ‘never’. For each food item, the nutrient content per average unit (specified serving size: slice, glass, or natural unit) was considered and women were asked how often they had used an amount of each food on average over the last year. Nutrient intakes were computed by multiplying the frequency response by the nutrient content of specified portion sizes using Microsoft^®^ Office Access 2007. The food composition database for calculating nutrient intakes took advantage of information from the US Department of Agriculture food composition tables [23] and it was complemented, when necessary, with a nutrient database developed by the National Institute of Nutrition in Mexico [24]. Total energy and nutrient intake were calculated by adding up the energy and nutrients from all foods. All the responses were converted to per day consumption.

To measure physical activity during the last year, a 7-day recall questionnaire was used to assess the participant’s physical activity (light-, moderate-, and vigorous intensity) during a usual week.

### 2.3. Dietary Scores

A summary of food components included in each diet score is presented in Table 1.

#### 2.3.1. DASH (Dietary Approaches to Stop Hypertension) Score

The DASH diet was designed to control blood pressure [7] and a score was constructed to measure adherence to this diet. It is based on 8 food components: intake of fruits, vegetables, legumes, fat and dairy products, whole grains, sodium, sweetened beverages, and red and processed meats. Each component of the score was divided into quintiles based on their intake for each participant. For fruits, vegetables, legumes, dairy products, and whole grains higher intakes were recommended. Therefore, women in the first quintile were assigned a score of 1 and women in the fifth quintile, a score of 5 points. For the remaining three components (sodium, red and processed meats, and sweetened beverages), low intake were recommended and therefore, the fifth quintile of intake was given a score of 1 point while the first quintile, received a score of 5 points. Finally, the overall DASH score was calculated by adding up the scores from each component. The DASH score ranged from 8 to 40, a higher score representing a higher adherence to the dietary recommendations to stop hypertension.

#### 2.3.2. HEI Score (The Healthy Eating Index)

The HEI was designed to assess diet quality of the Americans according to the US dietary guidelines [8]. The HEI score was computed using HEI-2015 which is made of 13 components: 9 adequacy components (total fruit, whole fruit, total vegetables, greens and beans, whole grains, dairy products, total protein foods, seafood and plant proteins, and fatty acids), and 4 moderation components (refined grains, sodium, added sugars and saturated fats) [25]. For the adequacy components, increasing levels of intake receive increasingly higher scores; whereas for the moderation components, increasing levels of intake receive decreasingly lower scores. The total HEI score ranged from 0 to 100 which was obtained by adding up the score of all the 13 components. The higher the score, the better was the diet quality.

#### 2.3.3. Alternate Mediterranean Diet Score (aMED)

The Mediterranean Diet Score was originally constructed from the traditional Mediterranean diet [26] and was later modified to include intake of sweets or sugar products [9]. In brief, the score was made of ten components and included 6 beneficial components (vegetables, fruits, cereals, legumes, fish and seafood, and ratio of monounsaturated lipids to saturated lipids) and four presumed detrimental dietary components (red meat, milk and dairy products, alcohol, and carbohydrates). The median consumption of each dietary component was calculated for every participant. A value of zero or one was assigned to the six food groups based on their intake level of below or above the median value, respectively. A value of one or zero was assigned to four potential harmful dietary components when the participant’s intake level was below or above the median value, respectively. The ten component scores were then summed resulting in a score ranging from 0 (minimal adherence to a Mediterranean diet) to 10 (maximum adherence).

#### 2.3.4. Diet Quality Index for Adults (DQI_A)

The DQI was developed as an instrument to assess the overall diet quality which reflects a risk gradient for major chronic diseases related to diet [10]. The DQI measure adherence to general dietary recommendations, including four components [27]: diet quality, diversity, equilibrium and physical activity. To calculate the three dietary components, the daily diet was divided into the following eight food groups: (1) water, (2) grains, (3) vegetables, (4) fruits, (5) milk products and cheese, (6) meat, fish, eggs and poultry (7) fat and oils, and (8) sweets, desserts, and snacks. For calculating the diet quality component, all the food items from the FFQ were subcategorized into groups based on their energy density and nutrient content. Dietary diversity was calculated by allocating one point to each of the eight food groups when at least one food item of that food group was consumed by the subject. The third component of the index, the dietary equilibrium score, was calculated by subtracting the excess score (percentage of intake of each main food group exceeding the upper level of its recommendation) from the adequacy score (percentage of the minimum recommended intake for each of the main food groups). The physical activity score, was obtained by dividing total minutes spent in moderate to vigorous physical activities per day by 30 min and multiplying with 100 to obtain a percentage expressing the compliance with the physical activity recommendations for adults. All components of the score were expressed as percentages and values were truncated at 100% when it exceeded 100%.

#### 2.3.5. Glycemic Index and Glycemic Load

Details of calculation of glycemic index (GI) and glycemic load (GL) are reported elsewhere [28]. The values of glycemic index for each food item were derived from the Foster–Powell tables [29]. All foods that contained carbohydrates were assigned GI values while food items that did not contain any carbohydrates were assigned a GI value of zero. The GL for each food item was calculated by multiplying the carbohydrate content of one serving by the food’s GI value. Therefore, each unit of GL represented the equivalent blood glucose-increasing effect of 1 g carbohydrates from white bread (or glucose depending on the reference used in determining the GI). The dietary GL was calculated by multiplying the available carbohydrate content of each food by its GI value and then multiplying the resultant value with the amount of consumption (divided by 100) and then summing the values from all food items. The overall GI was estimated by dividing the dietary GL by the total amount of consumed carbohydrates.

### 2.4. Statistical Analyses

Baseline characteristics of continuous variables were presented using means ± standard deviations (SD), while categorical variables were presented using percentages (%), overall and by categories of BMI (using WHO cut-off points of <25 kg/m^2^ as normal, 25–29.9 kg/m^2^ as overweight, and >=30 kg/m^2^ as obese). Differences in baseline characteristics by BMI categories were tested with analysis of variance (ANOVA) for continuous variables and chi-2 test for categorical variables.

The association of each dietary score modeled in tertiles with each anthropometric measurement (BMI, waist circumference, hip circumference and WHR) was examined using generalized linear regression models. For HEI, aMED, and DASH, the first tertile (corresponding to low diet quality) was used as reference while for GI and GL, the third tertile was used as reference. To examine the trends, the dietary score tertile variables were modeled as continuous variables. Socioeconomic status (low, medium, and high), educational level (no education, primary school, secondary school, and after school education), recruitment center (Mexico City, Monterrey, Veracruz), occupation (housewife and all other occupations), smoking status (never, former smoker and current smoker), alcohol consumption, marital status (married or living in a relationship and unmarried i.e., single or widow), use of oral contraceptive, hormone therapy, parity, number of full term pregnancies, breastfeeding, menopausal status, energy intake and physical activity were evaluated as potential confounders. Only socioeconomic status, recruitment center, education, energy intake and physical activity were retained in the final model.

Stratified analyses were conducted for factors known to be associated with obesity and/or dietary patterns, including physical activity (above or below the median), education (elementary or above), menopausal status (pre or post menopause), age (above or below the median), age (years) at first full-term pregnancy (<22, >=22) and number (N) of pregnancy (<3, >=3). Interactions were tested in the generalized linear model by adding an interaction term between the score and the stratification variable considered (in 2 levels).

All analyses were performed using Stata MP Version 14.1 (Stata Corporation, College Station, TX, USA) and IBM SPSS Statistics 20.0 (IBM Corp., Armonk, NY, USA). All P-values were two-sided and *p*-values < 0.05 were considered statistically significant.

## 3. Results

Baseline characteristics of the study population are summarized in Table 2. The average age at recruitment was 51.1 years and of the majority of the participants were from Mexico City (57%). Most of the women were married (68%), housewives (67%). and 94% were parous (of which 70% had 3 children or more). Sixty-seven percent were non-smokers and 76% non-alcohol drinkers while 56% were postmenopausal, of whom only 15% used menopausal hormone therapy (MHT).

The 1062 study participants were divided into three groups according to BMI categories, of whom 145 were normal weight (NW), 413 were overweight (OW), and 504 were obese (OB). Compared to normal weight women, overweight and obese women were generally older at recruitment and at menopause and younger at menarche. Obese women were also less educated than NW women, more frequently housewives, less physically active, and had more children.

Based on the different food items obtained from the FFQs, four dietary scores (“Dietary Approaches to Stop Hypertension”, “Mediterranean diet”, “Healthy Eating Index”, and “Diet Quality Index”) as well as glycemic index and glycemic load were constructed for each participant, as described above. The associations between these dietary scores/indexes and anthropometric measures, adjusted for age, education, center, energy intake, and physical activity, are presented in Table 3.

The aMED was statistically significantly associated with waist circumference: women in the 3rd tertile of aMED score had on average a 2.68 cm lower waist circumference compared to women in the 1st tertile (95% CI: −4.71; −0.65, Ptrend = 0.01). A higher adherence to aMED was also associated with a lower WHR (T3 versus T1 = −0.018; 95% CI, −0.03, 0.00; *p* = 0.02). aMED was not significantly associated with BMI or hip circumference.

Additionally, we found significant associations between GI and BMI (T1 versus T3 = −0.83, 95% CI, −1.64; −0.01; Ptrend = 0.05) and waist circumference (T1 versus T3 = −2.15; 95% CI, −4.29; −0.02, *p* = 0.05). No significant association was observed with GL nor with DASH, HEI, or DQI.

There was no statistically significant interaction in the associations between dietary scores and anthropometric measures with respect to age (median), menopausal status, physical activity (median), education, age at full-term pregnancy and number of full-term pregnancies. However, a statistically significant interaction with physical activity was observed in the associations between aMED and waist and WHR (P_interaction_ = 0.028 and 0.032, respectively). Women with a physical activity level above the median experienced a stronger decrease in waist circumference (β = −2.48; 95% CI, −3.97, −0.99 for one tertile increase in aMED; *p* = 0.001) and WHR (β = −1.52; 95% CI, −2.68, −0.36 for one tertile increase in aMED; *p* = 0.01), while no statistically significant change was observed among women with a physical activity level below the median (Figure 1).

## 4. Discussion

In this study we evaluated four dietary patterns, as well as GI and GL among Mexican women and their association with anthropometric measurements. Our results suggest that adherence to a Mediterranean diet is associated with a lower waist circumference and WHR. In addition, a high GI was positively associated with a higher BMI, and WC, while other dietary patterns and GL did not show any significant association with anthropometric indices in this population.

Several studies have reported a significant association between adherence to a priori dietary patterns and decreased body weight and central obesity [11]. The evidence of the association between aMED and obesity was reviewed by Garcia-Fernandez et al. [30]. A majority of cross-sectional studies reported similar results to our study and showed a strong inverse association between adherence to a Mediterranean diet and prevalence of obesity. This was confirmed in prospective cohort studies that showed that higher adherence to a Mediterranean diet was associated with a reduced weight gain and a lower risk of developing obesity. Results from the EPIC-PANACEA study were particularly consistent with our findings and showed a significant association of adherence to aMED with lower waist circumference in both men and women, while the Mediterranean diet was not associated with BMI [31]. Results from interventional dietary studies showed that adherence to a Mediterranean diet was associated with a lower BMI, and a significantly lower abdominal obesity [32,33,34]. All these studies were conducted in Europe and North-America and very little is known about other populations, and Mexican populations in particular. Our findings did not demonstrate any significant association of anthropometric measures with other common dietary patterns like HEI, DASH, and DQI in this Mexican population. In our study, we used the latest version of HEI which is designed to align with the 2015–2020 Dietary Guidelines for Americans (DGA). Regarding HEI, similar results were observed in a national survey conducted among Mexican Americans that showed no significant association between HEI and waist circumference among women, while the results were significant for men. DASH diet is broadly promoted for the prevention and treatment of high blood pressure and is considered as an example of a healthy eating pattern [35]. This diet was pilot tested in an interventional study of obese and overweight Latino adults residing in the United States and the investigators reported favorable results for weight loss and a change in BMI and suggested this approach could be useful for clinical weight loss programs in Latinos [36]. More recently, a study which examined adherence to DQI and anthropometric parameters reported that a mean increase in DQI was associated with a decrease in WC and BMI in men, while no longitudinal associations were found in women [37]. These gender-differences might be explained by different fat deposition in men and women with women tending to store fat in the lower extremities while men store fat in the abdominal region [38]. To our knowledge, one study has evaluated the association between a priori diet scores and anthropometry in Mexico [15]. This study included 3 different dietary scores: the cardioprotective index (CPI), the micronutrient adequacy index and the dietary diversity index (DDI), which have a larger emphasis on micronutrient intakes compared to the scores used in the present analysis. Based on DDI, there was a slightly lower diet diversity among normal weight participants compared to overweight and obese.

The four dietary patterns included in this study have many similarities and recommend consumption of fruits, vegetables, and whole grains, while they discourage excess consumption of red and processed meat. They also differ in many aspects, such that the Mediterranean diet has higher levels of monounsaturated fat and low saturated fat. It also contains lots of complex carbohydrates from legumes and adequate fiber from vegetables and fruits [39]. This diet being rich in dietary fiber provided by plant-based foods and having a low energy density and low glycemic load differentiates it from other dietary patterns. Beside these characteristics, high water content of this diet leads to increased satiation and low calorie intake, which in turn prevents weight gain [40], and this maybe the reason we observe association between Mediterranean diet and anthropometric measures.

It has been reported that consumption of foods with high glycemic index may increase risk of obesity, diabetes and cardiovascular diseases [41,42]. Moreover, a systematic review which assessed the effects of low glycemic index or load diets reported that overweight and obese people lost more weight and had an improved lipid profiles than those receiving diets with higher glycemic index and load [12]. In our study, GI but not GL was positively associated with higher anthropometric parameters. These findings are in line with the conclusions of another study conducted in Denmark, where high GI diets were strongly related to changes in body weight and WC in sedentary women, but not in men [43]. However, a recent prospective cohort study in a Mediterranean country like Spain did not find any consistent association between a higher GI and a higher weight gain [44].

Our study had some limitations. Study participants were part of the control group of a breast cancer case control study and therefore are not representative of the whole Mexican population, however they were population based controls. BMI classes were not evenly distributed, and the proportion of normal weight participants was much smaller than overweight and obese individuals, which might be the reason why dietary patterns had weak correlations with anthropometric parameters. Another limitation is the cross-sectional design of the study, which does not allow to draw conclusions about causal relationships. Also, the results were based on information assessed at one point in time and any within-person variation may lead to attenuation of the associations. Another shortcoming of cross-sectional analysis is that obese participants may have adopted a particular diet. Moreover, these individuals may be giving socially-desirable answers, and as such underestimate or under-report foods considered as unhealthy and overestimate beneficial foods like fruits and vegetables [45]. Energy intakes calculated from these FFQs are meant to rank individuals according their energy intake while not considering their absolute energy intakes. The closed nature of this FFQ did avoid extreme under- or over-reporting of energy intake, as such no misreports were identified. The fact that only one diet score was associated with anthropometry may indicate that, in our study population, these diet scores developed in Western populations are poor proxies of a healthy diet, particularly for scores with fixed cutpoints determined on other populations.

Nevertheless, this study also has some important strengths. We used a validated food-frequency questionnaire which was specifically designed for a Mexican population [22]. Few studies have evaluated diet quality using multiple dietary indices together and to our knowledge this has been the first study to evaluate diet quality of Mexican women using the 4 most common a priori indices together. Several factors such as SES, physical activity and smoking may confound the association of obesity and diet, but we had the possibility to control for all these factors.

## 5. Conclusions

In conclusion, adherence to aMED was associated with lower waist circumference and WHR and GI was positively associated with higher anthropometric indices. The other a priori dietary scores (DASH, HEI, DQI) appeared to be of limited value in exploring associations with anthropometric measures in Mexican women. Our study shows that a priori dietary patterns that were generally developed in populations from high income countries do not necessarily apply to populations from low and middle income countries. Other dietary scores based on local diet should be developed in order to provide recommendations that are adapted to the Mexican population.

## Figures and Tables

**Figure 1 nutrients-11-00603-f001:**
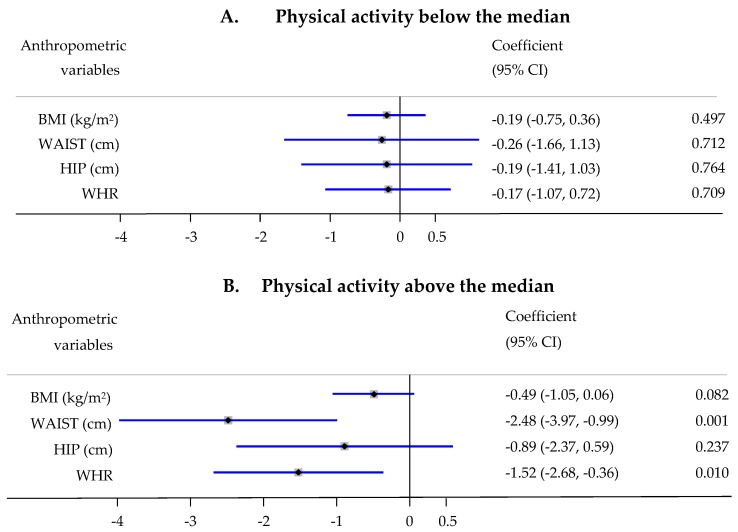
Association of Alternative Mediterranean Diet (aMED) score tertiles with anthropometric measurements stratified by physical activity—increase or decrease in 1 unit in anthropometric measurements (0.01 unit for WHR) per 1-tertile increase in the score, adjusted for age, center, education, and energy intake. A. Physical activity below the median; B. Physical activity above the median.

**Table 1 nutrients-11-00603-t001:** Comparison of food components included in each diet score—in green: high intake recommended; in orange: moderate intake recommended (only applicable for the Diet Quality Index (DQI)); in red: low intake recommended. DASH: Dietary Approaches to Stop Hypertension; HEI: Healthy Eating Index; aMED: Mediterranean Diet Score.

Food Groups	DASH	HEI	aMED	DQI
Fruits	Fruits and fruit juices	Total fruits	Fruits (including nuts)	Fresh fruits
		Whole fruits		Fruit juices, canned fruits, compote
				Fruit jam, fruit syrups
Vegetables	Vegetables (potatoes and legumes not included)	Total vegetables plus legumes	Vegetables (potatoes not included)	Fresh vegetables
				Canned vegetables or vegetables in sauces
				Vegetable burgers
Legumes	Nuts and legumes	Greens and beans	Legumes	Legumes (included in the group of meat replacements)
Grains, cereals and potatoes	Whole grains	Whole grains	Cereals	Whole grains and potatoes
		Refined grains		Refined grains and potato puree
		Plant protein		French fries, vienoiseries (e.g., croissants)
Animal products	Red and processed meat	Seafood	Red meat	Lean Meat
			Fish and seafood	Fish and egss
				Fatty meat
Dairy products	Fat and dairy products	Dairy products	Dairy products	Low fat Dairy products
				Full fat dairy products
Fats		Fatty acids ratio (MUFA+PUFA/SFA)	MUFA/SFA ratio	Vegetable (cooking fat) oil and margarine
				Butter
		Saturated fats		Cream, Mayonnaise and lard
Sodium and salty snacks	Sodium	Sodium		Salty snacks
Sweets	Sweetened beverages	Added sugar	Carbohydrates	Sweets, desserts, sweet snacks
Beverages				Water (beverages)
				Non-sugared and non-alcoholic beverages
			Alcohol	Alcohol and Sugar sweetened beverages
Other food groupings		Total protein food *		

* Total protein food includes: total meat, poultry, seafood (including organ meats and cured meats), eggs and legumes.

**Table 2 nutrients-11-00603-t002:** Baseline characteristics of the study population according to body mass index (*n* = 1062).

Variables ^1^	Overall (*n* = 1062)	BMI Classes			*p* Value ^6^
		Normoweight (*n* = 145)	Overweight (*n* = 413)	Obese (*n* = 504)	
*Mean* (*STD*)					
Age at recruitment (years)	51.1 (9.1)	49.2 (8.9)	50.6 (9.2)	52 (9.1)	0.002
Age at menarche (years), missing *n* = 16	12.9 (1.6)	13.2 (1.6)	12.9 (1.6)	12.7 (1.6)	0.002
Age at 1st full-term pregnancy (years) ^5^, missing *n* = 8	21.3 (4.7)	21.9 (4.5)	21.6 (4.7)	20.9 (4.9)	0.05
Age at menopause ^2^ (years), missing *n* = 17	47.1 (5.9)	45.2 (7.2)	47.7 (5.4)	47.2 (5.8)	0.01
Contraceptive use duration (years) ^3^, missing *n* = 84	3.9 (4.4)	3.4 (3.2)	3.8 (4.4)	4 (4.7)	0.69
Alcohol (g/day) ^4^	3.2 (7.4)	4.1 (6.6)	3.0 (6.8)	3.0 (8.2)	0.73
Energy intake (Kcal)	1867 (756)	1885 (768)	1848 (780)	1877 (733)	0.81
Weight (kg)	70.6 (13.6)	54.4 (6)	64.2 (5.9)	80.4 (12.2)	<0.001
Height (cm)	151.9 (6.3)	152.6 (6)	152.1 (6)	151.5 (6.5)	0.13
Body Mass Index (kg/m^2^)	30.5 (5.4)	23.3 (1.5)	27.7 (1.4)	34.9 (4.3)	<0.001
Waist circumference (cm), missing *n* = 10	99.4 (14)	85.6 (11.5)	94.4 (11.4)	107.4 (11.4)	<0.001
Hip circumference (cm), missing *n* = 8	109.3 (13.2)	97.3 (15.6)	103.3 (7.5)	117.7 (10.4)	<0.001
Waist to Hip ratio (WHR), missing *n* = 11	0.91 (0.1)	0.89 (0.13)	0.92 (0.12)	0.91 (0.07)	0.01
Total Physical activity (MET hours/week)	279 (51)	286 (63)	281 (50)	275 (48)	0.04
Light Physical Activity	196 (42)	191 (47)	194 (41)	199 (40)	0.06
Moderate Physical Activity	74 (76)	84 (91)	78 (77)	68 (70)	0.04
Vigorous Physical Activity	9 (35)	11 (46)	9 (33)	8 (34)	0.62
*N* (%)					
Center of recruitment					0.007
Mexico City	602 (57)	93 (64)	249 (60)	260 (52)	
Monterrey	261 (24)	26 (18)	87 (21)	148 (29)	
Veracruz	199 (19)	26 (18)	77 (19)	96 (19)	
Socioeconomic status					0.19
Low	355 (33.4)	50 (35)	140 (34)	165 (33)	
Medium	353 (33.3)	41 (28)	127 (31)	18 5 (37)	
High	354 (33.3)	54 (37)	146 (35)	154 (30)	
Education:					0.009
None	88 (8.3)	10 (6.9)	24 (5.8)	54 (10.7)	
Elementary	276 (26)	31 (21.4)	106 (25.7)	139 (27.6)	
Post Primary	322 (30.3)	47 (32.4)	120 (29.1)	155 (30.7)	
Secondary	269 (25.4)	35 (24.1)	111 (26.9)	123 (24.4)	
Vocational	62 (5.8)	11 (7.6)	30 (7.2)	21 (4.2)	
Professional	45 (4.2)	11 (7.6)	22 (5.3)	12 (2.4)	
Married	725 (68)	101 (70)	278 (67)	346 (69)	0.845
Occupation					0.001
Housewife	716 (67)	89 (61)	259 (63)	368 (73)	
Others	346 (33)	56 (39)	154 (37)	136 (27)	
Smoking status					0.27
Never	715 (67)	97 (67)	282 (68)	336 (67)	
Former	179 (17)	18 (12)	68 (17)	93 (18)	
Current	168 (16)	30 (21)	63 (15)	75 (15)	
Alcohol Consumers	250 (24)	35 (24)	106 (26)	109 (22)	0.35
Ever use of oral contraceptives	476 (45)	58 (40)	194 (47)	224 (44)	0.34
Parous	995 (94)	130 (90)	384 (93)	481 (95)	0.03
Number of full term pregnancies ^5^, missing *n* = 3					0.01
Number of FTP (1–2)	299 (30)	47 (36)	129 (34)	123 (26)	
Number of FTP (3+)	693 (70)	82 (64)	255 (66)	356 (74)	
Breast feeding ^5^					0.15
No breast feeding	101 (10)	14 (11)	38 (10)	49 (10)	
<12 months	226 (23)	31 (24)	102 (27)	93 (20)	
12+ months	668 (67)	85 (65)	244 (63)	339 (70)	
Menopausal status:					0.003
Postmenopausal	592 (56)	71 (49)	213 (52)	308 (61)	
Premenopausal	470 (44)	74 (51)	200 (48)	196 (39)	
Ever use of menopausal hormone therapy ^2^, missing *n* = 13	87 (15)	13 (19)	36 (17)	38 (13)	0.235

^1^ Number of missing values is 0 unless otherwise specified, ^2^ Among postmenopausal women only (*n* = 592), ^3^ Among oral contraceptive users only (*n* = 476), ^4^ Among alcohol consumers (*n* = 250), ^5^ Among parous women (*n* = 995), ^6^ Chi-square test or ANOVA to compare between BMI categories. MET=Metabolic Equivalent of Task.

**Table 3 nutrients-11-00603-t003:** Associations between a priori diet scores and anthropometric measurements, adjusted for age, education, center, energy intake, and physical activity.

Dietary Patterns	Body Mass Index (BMI)			Waist Circumference			Hip Circumference			WHR		
	Mean Difference	95% CI	*Ptrend* ^1^	Mean Difference	95% CI	*Ptrend* ^1^	Mean Difference	95% CI	*Ptrend* ^1^	Mean Difference	95% CI	*Ptrend* ^1^
**DASH**			*0.27*			*0.76*			*0.40*			*0.22*
T2 versus T1	−0.03	(−0.81; 0.76)		0.25	(−1.80; 2.31)		−0.44	(−2.36; 1.48)		0.006	(−0.01; 0.02)	
T3 versus T1	−0.52	(−1.42; 0.38)		0.37	(−1.99; 2.72)		−0.96	(−3.16; 1.25)		0.010	(−0.01; 0.03)	
**aMED**			*0.10*			***0.01***			*0.32*			***0.02***
T2 versus T1	−0.39	(−1.19; 0.40)		−0.79	(−2.87; 1.28)		−0.65	(−2.61; 1.30)		0.003	(−0.01; 0.02)	
T3 versus T1	−0.65	(−1.43; 0.13)		**−2.68**	**(−4.71; −0.65)**		−0.94	(−2.85; 0.96)		**−0.018**	**(−0.03; 0.00)**	
**HEI**			*0.66*			*0.37*			*0.10*			*0.55*
T2 versus T1	−0.02	(−0.82; 0.77)		−0.97	(−3.04; 1.09)		−0.92	(−2.86; 1.02)		0.001	(−0.01; 0.02)	
T3 versus T1	−0.19	(−1.00; 0.63)		−0.98	(−3.10; 1.15)		−1.68	(−3.67; 0.31)		0.005	(−0.01; 0.02)	
**DQI**			*0.58*			*0.31*			*0.27*			*0.76*
T2 versus T1	0.83	(0.02; 1.65)		1.67	(−0.46; 3.80)		1.86	(−0.13; 3.86)		−0.002	(−0.02; 0.01)	
T3 versus T1	0.26	(−0.58; 1.10)		1.17	(−1.01; 3.36)		1.19	(−0.85; 3.24)		0.002	(−0.01; 0.02)	
**GI**			***0.05***			***0.05***			*0.22*			*0.45*
T2 versus T3	−0.10	(−0.89; 0.69)		−0.21	(−2.26; 1.84)		−0.86	(−2.78; 1.07)		0.008	(−0.01; 0.02)	
T1 versus T3	**−0.83**	**(−1.64; −0.01)**		**−2.15**	**(−4.29; −0.02)**		−1.24	(−3.24; 0.75)		−0.006	(−0.02; 0.01)	
**GL**			*0.38*			*0.25*			*0.10*			*0.94*
T2 versus T3	−0.04	(−1.00; 0.92)		0.29	(−2.21; 2.79)		−0.90	(−3.23; 1.44)		0.006	(−0.01; 0.02)	
T1 versus T3	−0.51	(−1.72; 0.70)		−1.67	(−4.83; 1.49)		−2.40	(−5.36; 0.56)		0.000	(−0.02; 0.02)	

^1^*p* for trend based on tertile variable (per 1-tertile increase in the score). In bold: results that are statistically significant. DASH = Dietary Approaches to Stop Hypertension; aMED = Alternate Mediterranean Diet Score; HEI = Healthy Eating Index; DQI = Diet Quality Index for Adults; GI = Glycemic Index; GL = Glycemic Load; WHR = Waist-to-Hip Ratio.

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
