# Peer review of "Association of a Priori-Defined Dietary Patterns with Anthropometric Measurements: A Cross-Sectional Study in Mexican Women"

_nutrients, 2019, doi:10.3390/nu11030603_

Round 1
Reviewer 1 Report
This is an interesting article making good use of secondary analysis.
I would suggest authors consider redrafting section starting on line 136: "Therefore, women in the 1st quintile were 136 assigned a score of 1 and women in the 5th quintile, a score of 5 points. For the remaining three 137 components (sodium, red and processed meats, and sweetened beverages), low intake were 138 recommended and therefore, the highest quintile of intake was given a score of 1 point while the lowest quintile, received a score of 5 points". This is because there is potential for this to misunderstood. given the use of 1st vs 5th and highest vs lowest. Please use consistent language.
Generally the limitations are well documented but given the majority of women were older, and overweight/obese this may explain the null findings for the majority of issues though it is somewhat surprising that only one dietary score demonstrated any significant association given that the underlying principles of the scores are similar. It may be however, that this simply demonstrates that dietary indices are poor proxies and that the same dietary intake can be healthier in one index and less healthy in another in the same manner as an individual could be in the highest quintile in one cohort and lowest in another given the manner in which the indices are calculated.. This would explain the variation in findings across the studies and warrants further consideration/ discussion.
Author Response
Reviewer 1:
This is an interesting article making good use of secondary analysis.
I would suggest authors consider redrafting section starting on line 136: "Therefore, women in the 1st quintile were 136 assigned a score of 1 and women in the 5th quintile, a score of 5 points. For the remaining three 137 components (sodium, red and processed meats, and sweetened beverages), low intake were 138 recommended and therefore, the highest quintile of intake was given a score of 1 point while the lowest quintile, received a score of 5 points". This is because there is potential for this to misunderstood. given the use of 1st vs 5th and highest vs lowest. Please use consistent language.
This has now been rephrased.
Generally the limitations are well documented but given the majority of women were older, and overweight/obese this may explain the null findings for the majority of issues though it is somewhat surprising that only one dietary score demonstrated any significant association given that the underlying principles of the scores are similar. It may be however, that this simply demonstrates that dietary indices are poor proxies and that the same dietary intake can be healthier in one index and less healthy in another in the same manner as an individual could be in the highest quintile in one cohort and lowest in another given the manner in which the indices are calculated.. This would explain the variation in findings across the studies and warrants further consideration/ discussion.
We thank the reviewer for this suggestion. The following sentence has been added to the discussion: “The fact that only one diet score was associated with anthropometry may indicate that, in our study population, these diet scores are poor proxies of a healthy diet, in particular for scores with fixed cutpoints determined on other populations.”
Reviewer 2 Report
This study used a validated FFQ specifically designed for a Mexican population and comprehensively evaluated diet quality of Mexican women using a variety of indices, including DASH, HEI, MDS, DQI, GI & GL.
Overall, the study was properly designed and technical sound. Results and discussion were well presented in a clear and concise way that the readers can easily follow. Although the study had some limitations in terms of the study subjects, the findings of this study will still be useful to better understand the association of a priori-defined dietary patterns with anthropometric measurements in Mexican women.
I recommend this manuscript to be considered for publication after a few minor text editing.
Author Response
Reviewer 2:
This study used a validated FFQ specifically designed for a Mexican population and comprehensively evaluated diet quality of Mexican women using a variety of indices, including DASH, HEI, MDS, DQI, GI & GL.
Overall, the study was properly designed and technical sound. Results and discussion were well presented in a clear and concise way that the readers can easily follow. Although the study had some limitations in terms of the study subjects, the findings of this study will still be useful to better understand the association of a priori-defined dietary patterns with anthropometric measurements in Mexican women.
I recommend this manuscript to be considered for publication after a few minor text editing.
We thank the reviewer for his/her positive comments.
Reviewer 3 Report
This study describes the association with anthropometric markers of various a priori dietary scores in Mexico. It is of interest as it appears to be one of the first study of the kind, but I identified a number of issues that need to be addressed for this paper to be considered for publication.
Major general points: The statistical analysis should be revised and explained in greater detail. The association with fruit groups is never described in the objectives, neither in the statistical analysis nor in the results, but appears in the abstract. The cross-sectional design of the study should appear in the title, abstract and as a clear limitation that should be discussed.
Specifict points:
Abstract: legume intake is not in the objectives, sounds out of place and seems to defeat the purpose of dietary scores
Intro: 650 million (13%) of them is problematic, because it is 13% in total, not of them. Please rephrase
Ref (2): which year?
l.40: overall diet *quality*
l.46: in which time period?
l.48 a priori *scores* to avoid patterns repetition + give refs of the indices listed
l.55 define HIC
l.71: what were the results of the studies 11 and 12? What is the added value of this study? State clearly the objective emphasizing the unique features / novel character of your study
l.98: give more details on the questionnaires used, if validated or not, and how were the variables categorized
How was sodium intake estimated? It is well known that is is particularly hard to estimate from FFQs. Was any correction applied?
Did you exclude under-reporters?
l.132: the DASH score was not constructed to control blood pressure. Rather, the DASH diet, and various scores of adherence have been proposed. One of them is the one proposed by Fung, that you describe.
l.143: Description of the HEI could be largely shortened
l.170: As for DASH, there are plenty of versions of “Mediterranean diet scores” and the one you describe is only one of them, usually termed “aMED”, not “MDS”.
l.189: what is “rest group”
All scores specificities should be summarized in a table and the text should be much shortened.
l.234: the examination of the trend (linear?) should be described in greater detail. What was modelled as continuous, the tertile variables (one technique to assess the presence of linear trend)? Other ways would be the use of splines.
l.236 “recruitment center (list them)” !!!
l.243: on what grounds did you test the interactions with the 5 factors listed? Please give rationale for this, as performing a large amount of interaction tests and subgroup analyses can give way to spurious associations, as well as issues of statistical power in the subgroups with a small number of participants.
p-values one or two sided?
Table 1 could be reduced by only showing one line for all the binary variables, e.g. married, using oral contraceptives, current alcohol intake, etc. How is “alcohol intake” defined? Also, age at 1st full term pregnancy , indicate the actual sample size as there are 67 nulliparous women. Same for age at menopause (hald of the sample is premenopausal). Use of oral contraceptives: is it current or ever?
L.261-269: paragraph could be synthesized and numbers can be seen in the table, therefore don´t need to be repeated.
Figure 1 should go in supplemental. Beverages is not a food group per se, and should probably be taken out of the Figure 1. Moreover, the labels should be changed.
Instead or presenting the food groups, it would probably better to describe the diet scores.
Table 2 and results. It is very unclear from the table how were the tertiles modelled. Suggest a presentation on a figure as this table is pretty dense and hard to catch at a glance which scores show associations, with which measurements. Why are GI and GL modelled in quartiles?
Discussion l.310-329: why are you describing only a handful of studies that looked at Med diet in relation to weight (omitting a large number of studies from the literature) and how does it relate to your own results?
l.345 where are the data on the correlations?
In general, the discussion would benefit from more focus, comparison with other Mexican studies and put things in a middle income country focus.
Conclusion: the association between MDS and WC was not particularly strong. Please avoid the word “predict” in a cross-sectional study.
Author Response
Reviewer 3:
This study describes the association with anthropometric markers of various a priori dietary scores in Mexico. It is of interest as it appears to be one of the first study of the kind, but I identified a number of issues that need to be addressed for this paper to be considered for publication.
Major general points: The statistical analysis should be revised and explained in greater detail. The association with fruit groups is never described in the objectives, neither in the statistical analysis nor in the results, but appears in the abstract. The cross-sectional design of the study should appear in the title, abstract and as a clear limitation that should be discussed.
We have added the study design in the title and in the abstract and have expanded the discussion on the limitations of the cross-sectional design. We have now removed the food group results from the abstract.
Specific points:
Abstract: legume intake is not in the objectives, sounds out of place and seems to defeat the purpose of dietary scores
This result has now been removed from the abstract.
Intro: 650 million (13%) of them is problematic, because it is 13% in total, not of them. Please rephrase
Done
Ref (2): which year?
In 2012. This has been added to the text.
l.40: overall diet *quality*
Fixed
l.46: in which time period?
Between 1990 and 2010. This has been added to the text.
l.48 a priori *scores* to avoid patterns repetition + give refs of the indices listed
Done.
l.55 define HIC
High Income Countries. This has been added to the text.
l.71: what were the results of the studies 11 and 12? What is the added value of this study? State clearly the objective emphasizing the unique features / novel character of your study
In fact only one study (ref 12) examined the association between a priori dietary scores and anthropometry in the Mexican population. This unique study used different a priori scores (a cardioprotective index; a micronutrient adequacy index; and a dietary diversity index). Our study is therefore the first to explore DASH, HEI, aMED and DQI in the Mexican population. The text has been modified accordingly.
l.98: give more details on the questionnaires used, if validated or not, and how were the variables categorized
How was sodium intake estimated? It is well known that is is particularly hard to estimate from FFQs. Was any correction applied?
Did you exclude under-reporters?
The FFQ was validated. Other questionnaires were not but have been described extensively in previous publications from the CAMA study. References to these publications have now been added to the text.
Only non-discretionary salt was considered for the calculation of the sodium intakes. Sodium intake has been estimated as other nutrients, with no specific correction. Nutrient intakes were computed by multiplying the frequency response by the nutrient content of specified portion sizes using a program developed at the National Institute of Public Health in Mexico. The database for calculating the nutrient intakes used information from the U.S. Department of Agriculture food composition tables (Department of Agriculture. Composition of foods: raw, processed, prepared, 1963-1991. Agricultural handbook. Washington (DC): Government Printing Office; 1992.) complemented when necessary by a nutrient database developed by the Mexican National Institute of Nutrition (Chavez N, Hernandez M, Roldan J, editors. Valor nutritivo de los alimentos de mayor consumo en Me´xico. Mexico City, Mexico: Comisio´n Nacional de Alimentacio´n y Instituto Nacional de la Nutricio´n; 1992.)
Under-reporters were not excluded. We have added a comment on that in the discussion.
l.132: the DASH score was not constructed to control blood pressure. Rather, the DASH diet, and various scores of adherence have been proposed. One of them is the one proposed by Fung, that you describe.
This has now been clarified.
l.143: Description of the HEI could be largely shortened
This paragraph has now been shortened.
l.170: As for DASH, there are plenty of versions of “Mediterranean diet scores” and the one you describe is only one of them, usually termed “aMED”, not “MDS”.
MDS has been replaced by aMED throughout the text.
l.189: what is “rest group”
The rest-group includes all energy-dense and low nutritious foods that are not essential in a healthy diet and for which the consumption is recommended to be limited. This includes for instance sweets, desserts, snacks. The text has now been modified.
All scores specificities should be summarized in a table and the text should be much shortened.
We have now shortened the text and added a table presenting the food components included in each score.
l.234: the examination of the trend (linear?) should be described in greater detail. What was modelled as continuous, the tertile variables (one technique to assess the presence of linear trend)? Other ways would be the use of splines.
The tertile variables were modelled as continuous. This has now been added to the text.
l.236 “recruitment center (list them)” !!!
Fixed
l.243: on what grounds did you test the interactions with the 5 factors listed? Please give rationale for this, as performing a large amount of interaction tests and subgroup analyses can give way to spurious associations, as well as issues of statistical power in the subgroups with a small number of participants.
Stratified analyses and interaction tests were conducted for factors known to be associated with obesity and/or dietary patterns. This has been added.
p-values one or two sided?
Two-sided. This has been added to the text.
Table 1 could be reduced by only showing one line for all the binary variables, e.g. married, using oral contraceptives, current alcohol intake, etc. How is “alcohol intake” defined? Also, age at 1st full term pregnancy , indicate the actual sample size as there are 67 nulliparous women. Same for age at menopause (hald of the sample is premenopausal). Use of oral contraceptives: is it current or ever?
Table 1 has now been reduced as suggested and sample sizes have been added in the footnotes.
L.261-269: paragraph could be synthesized and numbers can be seen in the table, therefore don´t need to be repeated.
Numbers have been deleted from the text.
Figure 1 should go in supplemental. Beverages is not a food group per se, and should probably be taken out of the Figure 1. Moreover, the labels should be changed.
Instead or presenting the food groups, it would probably better to describe the diet scores.
As rightly pointed out by the reviewer, because the manuscript is focused on dietary patterns and not food groups, we have now decided to remove figure 1 and all analyses on food groups.
Table 2 and results. It is very unclear from the table how were the tertiles modelled. Suggest a presentation on a figure as this table is pretty dense and hard to catch at a glance which scores show associations, with which measurements.
We have now simplified the presentation of table 2 by including only the results based on tertiles.
Why are GI and GL modelled in quartiles?
This was a typo and has been corrected, GI and GL were modelled in tertiles and not quartiles.
Discussion l.310-329: why are you describing only a handful of studies that looked at Med diet in relation to weight (omitting a large number of studies from the literature) and how does it relate to your own results?
To be more comprehensive, we have now based our description of existing literature on Mediterranean diet and obesity on the review paper by Garcia-Fernandez et al.
l.345 where are the data on the correlations?
What we meant here is that the dietary patterns have many similarities. This has been re-phrased.
In general, the discussion would benefit from more focus, comparison with other Mexican studies and put things in a middle income country focus.
To our knowledge, there is only one other Mexican study that examined the association between a priori dietary patterns and obesity. We are now providing more information on this study in the discussion.
Conclusion: the association between MDS and WC was not particularly strong. Please avoid the word “predict” in a cross-sectional study.
This has been corrected.
Round 2
Reviewer 3 Report
I thank the authors for providing mostly satisfying answers and modifying the manuscript, which I think have improved a lot. However, I have the following remaining points of concern:
- It is not very common to display a B coefficient per 1-tertile increase. I would like to see the mean difference of anthropometric marker in each of the tertile taking one as a reference, or provide the B for 1 SD increase in the dietary score (using the score as a continuous independent variable).
- Adjustment of the main results is limited (age, center, education) and the reason to retain only covariates that produce a change in the coefficient >10% is debatable. Including sex, energy intake and physical activity would seem to be necessary. In particular, adjusting for energy intake is important to have an idea of the association of the diet quality beyond the quantity.
- I really appreciate the table with the summary characteristics of the scores, but I feel it could be improved, to state at least if the components are deemed beneficial or consumption should be limited, and have a left column with the overarching food group.
Author Response
Point-by-point response to reviewers:
Reviewer 3:
I thank the authors for providing mostly satisfying answers and modifying the manuscript, which I think have improved a lot. However, I have the following remaining points of concern:
- It is not very common to display a B coefficient per 1-tertile increase. I would like to see the mean difference of anthropometric marker in each of the tertile taking one as a reference, or provide the B for 1 SD increase in the dietary score (using the score as a continuous independent variable).
We are now presenting mean differences in each tertile in table 2.
- Adjustment of the main results is limited (age, center, education) and the reason to retain only covariates that produce a change in the coefficient >10% is debatable. Including sex, energy intake and physical activity would seem to be necessary. In particular, adjusting for energy intake is important to have an idea of the association of the diet quality beyond the quantity.
Results are now adjusted for energy intake and physical activity in addition to age, center and education (sex was not considered as only women are included).
- I really appreciate the table with the summary characteristics of the scores, but I feel it could be improved, to state at least if the components are deemed beneficial or consumption should be limited, and have a left column with the overarching food group.
We have now added a colour code to indicate food components with high or low intake recommended and a left column with the overarching food group.